# Task Assignment Algorithm Based on Trust in Volunteer Computing Platforms

**Ling Xu [1,2,*], Jianzhong Qiao [1], Shukuan Lin [1] and Ruihua Qi [2]**

[1] School of Computer Science and Engineering, Northeastern University, Shenyang 110819, China
[2] School of Software Engineering, Dalian University of Foreign Languages, Dalian 116044, China
* Correspondence: xuling@dlufl.edu.cn; Tel.: +86-158-4097-5069

**Abstract:** In volunteer computing (VC), the expected availability time and the actual availability time provided by volunteer nodes (VNs) are usually inconsistent. Scheduling tasks with precedence constraints in VC under this situation is a new challenge. In this paper, we propose two novel task assignment algorithms to minimize completion time (makespan) by a flexible task assignment. Firstly, this paper proposes a reliability model, which uses a simple fuzzy model to predict the time interval provided by a VN. This reliability model can reduce inconsistencies between the expected availability time and actual availability time. Secondly, based on the reliability model, this paper proposes an algorithm called EFTT (Earliest Finish Task based on Trust, EFTT), which can minimize makespan. However, EFTT may induce resource waste in task assignment. To make full use of computing resources and reduce task segmentation rate, an algorithm IEFTT (improved earliest finish task based on trust, IEFTT) is further proposed. Finally, experimental results verify the efficiency of the proposed algorithms.

**Keywords:** volunteer computing; task assignment; availability; reliability model

## 1. Introduction

In the past decades, volunteer computing (VC) has provided huge computing power for large-scale scientific research projects by using idle resources over the Internet. A well-known open volunteer computing platform (VCP) is BOINC [1] (Berkeley Open Infrastructure for Network Computing), and some scientific research projects are running in BOINC, such as SETI@home [2] and Einstein@ Home. In addition, there are also some *VC* projects running in other VC platforms, such as Folding@home [3] and ATLAS@Home [4]. The network structure of these volunteer computing platforms (VCPs) is usually a master-worker distributed network computing model [5], as is shown in Figure 1. In Figure 1, the computers that provide computing resources are called volunteer nodes (VNs) and the server is responsible for assigning tasks and recycling results.

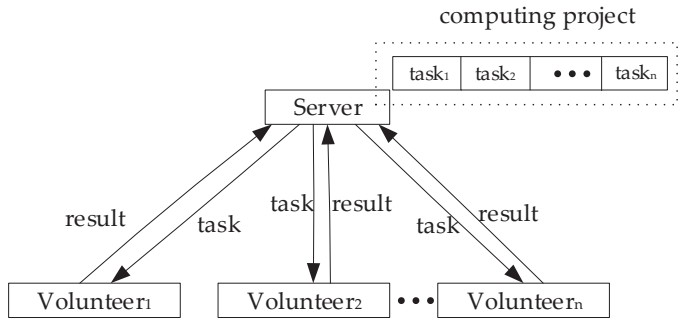

**Figure 1.** Master-worker model.



As VC is recognized for high-throughput computing, more and more scientific problems have been deployed in VCPs. Because the donated resources of VN are not available all the time, the completion time (makspan) of a task is usually hours or days. Therefore, the goal of many task assignment algorithms in prior studies [6–9] was to maximize the rate of task completion. However, as mentioned in References [10,11], minimizing the turnaround time of a task or batches of tasks was very meaningful for some special VC applications. Moreover, VC Applications such as SETI@home should be more efficient if they use minimizing a turnaround as the goal of task assignments. However, there are few studies about makespan in VC and the prior studies did not pay much attention to the reliability in task assignment.

Reliability is often associated with fault tolerance and fault tolerance is often associated with task failure or malfunctioning workers. Task failure or malfunctioning workers can be solved by redundant tasks or other techniques such as rescheduling [12]. Although these techniques are helpful for reliable task assignment, the performance of scheduling would be improved if reliability is taken into account in task assignment.

Usually, the reliability of the distributed systems is defined as the probability that the task can be executed successfully [13]. In the current context for task assignment, the definition can be narrowed down to the gap between the expected availability time and actual availability time. The gap characterizes the performance of task assignment, and the smaller the gap, the better the performance of task assignment. Because VC differs from other distributed computing such as grid computing and cloud computing. In VC, the expected availability time of VN is often disturbed by some reasons, such as the non-VC CPU load exceeding a threshold or another purpose. In such circumstances, there exists a gap between the actual makespan and the expected makespan, which causes the performance degradation of task assignment. Consequently, reliability is very useful for the task assignment of VCPs.

Moreover, many prior studies [7,9,14] about task assignment focused on independent tasks. However, the tasks of some VC applications in practice are precedence-constrained. Generally speaking, task priority can be represented by DAG (directed acyclic graph) [15], where nodes denote tasks and each edge $(t_i \rightarrow t_j)$ denote task priority (i.e., task $t_j$ cannot be executed until task $t_i$ has been completed). For example, as is shown in Figure 2, task $t_4$ cannot be executed until $t_1$ and $t_2$ have been completed. To facilitate the discussion of the task priority, a hypothetical root node is defined for the DAG. A hypothetical root node is a start node, whose computation time is 0.

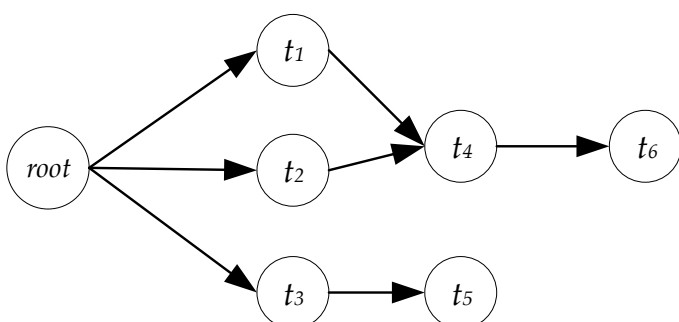

**Figure 2.** An example of the directed acrylic graph (DAG) with six tasks.

In this paper, we address the task assignment problem in VCPs and propose two task assignment algorithms that take into account reliability, makespan and tasks with precedence-constrained. The main contributions of this paper are summarized as follows:

(1) We propose a reliability model to reduce the inconsistencies between the expected availability time and actual availability time in VCPs.

(2) We propose two novel algorithms on the basis of the reliability model, that is, EFTT algorithm (earliest finish task based on trust, EFTT) and IEFTT algorithm (improved earliest finish task based

on trust, IEFTT). Unlike the existing algorithms, these two algorithms can minimize the makespan by considering reliability and task priority.

(3) We conduct simulation experiments to evaluate the efficiency of the proposed algorithms. The experimental results show that our proposed algorithms are more efficient than the existing algorithms.

The remainder of this paper is organized as follows: in the next section, we review the related work. Section 3 introduces the definition of the problems. Section 4 illustrates our task assignment algorithms. Section 5 gives the experimental results and analysis of the proposed task scheduling algorithms. Section 6 concludes this paper.

## 2. Related Work

In this section, we summarize the related work of task assignment in VCPs. To more comprehensively describe the related work of task assignment, this paper introduces the related work in two ways; one is a task assignment in distributed computing and the other is a task assignment in volunteer computing.

### 2.1. Task Assignment in Other Distributed Computing Systems

How to assign parallel tasks to multiple computers has been extensively studied, such as HEFT (Heterogeneous Earliest Finish Time, HEFT) [16], Min-min [17], and Max-min [18] . These prior studies use DAG to assign tasks according to task priority in static task assignment. Moreover, there are also other studies [19,20] about dynamic task assignment with task priority. However, the central issue in these studies is that they did not take into account the reliability.

Some studies that paid attention to reliability in task assignment, such as the studies in References [21,22]. These studies adopt search based on state space to obtain the optimal solution in task assignment. However, these methods have high computational complexity. Therefore, some studies adopted the heuristic algorithm to reduce computational complexity. Kang et al. [23] proposed an iterated greedy algorithm to maximize reliability in task assignment. They adopted stochastic local search to reduce the computational complexity. Furthermore, a dynamic resource assignment MOC algorithm [24] for an independent task was proposed by Salehi. The objective of the algorithm is to maximize the number of tasks completed while ensuring reliability. Specifically, the MOC algorithm defines a stochastic robustness to assign tasks and discards tasks that miss their deadlines to maximize the number of completed tasks. However, these studies are different from our work, because they did not consider task priority in task assignment.

Moreover, there are also some task assignment algorithms for specific application scenarios, such as cloud computing [25], mobile social networks [26] and grid computing [27] et al. Because these scenarios are different from volunteer computing where volunteer nodes may be often disconnected because of some reasons, such as the computer is in use, or the non-VC CPU load exceeds a threshold, the task assignment algorithm for such scenarios cannot be directly applied to the VCPs.

### 2.2. Task Assignment in Volunteer Computing Platforms

Effective task assignment can improve the system performance, so many task assignment algorithms have been extensively studied, such as the studies in References [28–30], which mainly take into account the attributes of tasks and resources constraints, such as the deadline and size of the tasks. However, these studies did not take into account that the VN is not always available. The study proposed in Reference [31] shows that the VNs will not be always available all the time according to the analysis of large scale real-world records in VCPs. Moreover, the expected availability time provided by a volunteer node will be interrupted by some reasons as mentioned before, which will make the turnaround of applications longer.

In this paper, we adopt a reliability model to reduce the disturbance caused by some reasons. In our reliability model, we associate a trust value with each VN to predict the availability time

of each VN. Usually, trust value is defined as the credibility of each VN. In the context of this paper, we define the trust value as the credibility that a VN can provide the expected availability time. Moreover, the study in Reference [32] shows that the higher trust value of the VN, the closer its expected availability time and the actual availability time are, so we associate a trust value with each VN in our reliability model to reduce the disturbance and improve the performance in task assignment.

The closest work to our study is two novel task scheduling algorithms proposed by Essafi et al. [32]. These two algorithms are called the HEFT-AC algorithm, which mainly modifies the HEFT algorithm to make it suitable for the volunteer computing platforms and the HEFT-ACU algorithm associates the reputation value of each VN to improve reliability. However, the HEFT-AC algorithm and the HEFT-ACU algorithm are mainly about the assignment of independent tasks, which are different from our work. In our work, we mainly focus on the task assignment that meets task priority requirement in VCP and the objective is to minimize the total completion time (makespan) on the basis of improving the reliability by associating trust value of each VN.

## 3. Task Assignment Problem

### 3.1. Problem Description

Since VN is not always available over time, this paper associates the trust value of each VN to establish a reliability model that aims at reducing the disturbances caused by some reasons as mentioned before, and minimizing the makespan. The notations used in this paper are summarized in Table 1.

**Table 1.** Notations used in problem description.

| Notation | Notation Meaning |
|---|---|
| $T$ | The set of tasks |
| $t_i$ | The $i$th task of the set $T$ |
| $V$ | The set of volunteer nodes |
| $v_j$ | The $j$th volunteer node of the set $V$ |
| $|V|$ | The number of nodes in the set $V$ |
| $h_j$ | The node $v_j$ contributes $h_j$ hours |
| $v_j.trust$ | The trust value of the node $v_j$ |
| $t_i.cost$ | The completion time needed of $t_i$ |
| $Range_j[l_a, l_b]$ | The confidence interval of the node $v_j$ |
| $ratio_v$ | the ratio of the actual availability time of node $v$ to its expected availability time |
| $probability_v$ | the probability of different ratios |
| $D(v_j)$ | the total time assigned to the node $v_j$ |

Although VC applications currently have no direct support for dividing long variable-sized tasks into small-sized tasks, many studies have integrated VC with other distributed computing to meet their special requirement, such as the study in Reference [33], which discussed the strategies about how to divide long-running GARLI analyses into short BOINC workunits (a unit of work in the BOINC platform). On this basis, to extend the VC, in this paper, we assume tasks of a VC application are arbitrarily divisible.

Given a set of tasks in *VCPs*, denoted by the set $T = \{t_1, t_2, \ldots, t_m\}$. Suppose that each task $t_i$ can be completed by any node $v_j$ or several other nodes in *VCPs*. In addition, we assume that the volunteer nodes provide the same computing power to different tasks and a set of volunteer nodes are denoted by the set $V = \{v_1, v_2, \ldots, v_2\}$. Moreover, we use the DAG known in advance to represent the interdependency of tasks. These are different from our prior work [34], which mainly focuses on how to assign independent tasks within deadline and the objective of task assignment is maximizing the number of tasks completed. Although the two papers take into account the dynamic of VC, this paper mainly uses a reliability model to predict the availability time to reduce the impact of dynamic, unlike our prior work using monitoring mechanism to achieve dynamic task scheduling.

The related concepts are given as follows:

**Definition 1 (task).** *Task $t_i$ is a double dimension array that is denoted by $(id, t_i.cost)$. The id represents the order of the tasks arrived; $t_i.cost$ represents completion time and the unit of $t_i.cost$ s an hour. At the same time, the cost of the task is only related to the number of task fragments, and it can be obtained by sampling.*

For example, as shown in Figure 3a, $t_1.cost = 2$, which means that the node $v_j$ will take two hours to complete the task $t_1$

**Definition 2 (prerequisite task).** *Given a task set, denoted by the set $T = t_1, t_2, \ldots, t_m$, and its corresponding DAG. When $t_j$ can be executed only after task $t_i$ has been completed, then task $t_i$ is called the prerequisite task of task $t_j$, and task $t_j$ is the post-order task of task $t_i$.*

For example, as shown in Figure 2, task $t_4$ can be executed only after task $t_1$ and task $t_2$ have been completed, so we call task $t_1$ and task $t_2$ are the prerequisite task of task $t_4$.

**Definition 3 (node).** *Given a volunteer node set, denoted by the set $V = \{v_1, v, \ldots, v_j\}$, and each node $v_j$ is a double dimension array, which is denoted by $(h_j, v_j.trust)$. The $h_j$ represents the expected availability time donated by the node $v_j$ is $h_j$ hours, and $v_j.trust$ represents the trust value that is the node $v_j$ can donate availability time which is $h_j$ hours. The parameter $v_j.trust$ of a node in this paper can be calculated by its historical records. In the future, we intend to study in depth the accurate calculation of parameter $v_j.trust$.*

For example, as shown in Figure 3b, $h_2 = 4$, $v_2.trust = 30\%$, which means the availability time donated by node $v_2$ is four hours and the trust value of this availability time is 30%.

| $T$ | $t_i.cost$ |
|-----|--------|
| $t_1$ | 2 |
| $t_2$ | 1 |
| $t_3$ | 3 |
| $t_4$ | 4 |
| $t_5$ | 2 |
| $t_6$ | 3 |

(a)

| $V$ | $h_j$ | $v_j.trust$ |
|-----|-------|--------|
| $v_1$ | 4 | 70% |
| $v_2$ | 4 | 30% |
| $v_3$ | 3 | 60% |
| $v_4$ | 4 | 50% |
| $v_5$ | 3 | 40% |

(b)

**Figure 3.** Task set and volunteer node set at time $l_1$. (**a**) task set at time $l_1$; (**b**) volunteer node set at time $l_1$.

VN may often be disconnected as mentioned before, so the expected availability time is different from the actual availability time. As shown in Figure 4, there is a difference between the expected availability time and the actual availability time provided by the node $v_2$. To reduce the gap between these values, this paper associates a trust value with each volunteer node and builds a reliability model.

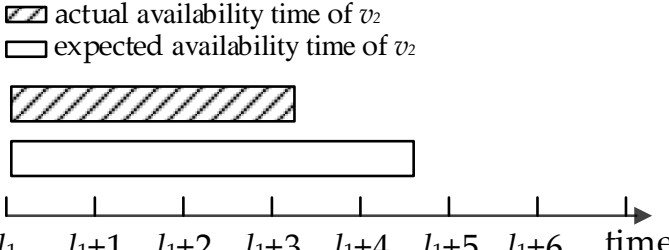

**Figure 4.** The expected availability time and the actual availability time of the node $v_2$.

## 3.2. Reliability Model

The computational resources donated by VNs can be represented by some scattered time intervals [31]. Based on this viewpoint, this paper defines the confidence interval of the node in the reliability model.

**Definition 4** (**confidence interval**). *The confidence interval of the node $v_j$ at time $l_1$ is denoted by $Range_j[l_a, l_b]$, which means that the node $v_j$ can keep the state of donating resources before time $l_a$ and disconnect at any time in the time interval $[l_a, l_b]$.*

The confidence interval is very important for both the EFTT algorithm and IEFTT algorithm. Specifically, we estimate the confidence interval of each VN by a simple fuzzy model according to the relationship between actual availability time and expected availability time in the historical record. Although the availability time of a node may be online longer than what it originally claimed, we only assign tasks to a node within its expected availability time. Therefore, in our algorithms, the upper bound "Range" is the expected online time. Because a node is not often available within its expected availability time, it is very hard to make sure the availability extra online time. In future, we will take into account how to assign tasks to extra online time.

The lower bound of "Range" is computed by the interpolation method. Specifically, we analyze the actual availability time and expected availability time of a node $v$ for nearly 100 times. After the statistical analysis of these values, we obtained the probability that different ratios of the actual availability time and expected availability time of node $v$ appeared is shown in Figure 5a. In Figure 5a, $ratio_v$ represents the ratio of the actual availability time of node $v$ to the expected availability time and the $probability_v$ represents the probability of different ratios. Therefore, for any expected availability time provided by node $v$ we can calculate the probability of node different actual availability time appeared by interpolation according to the statistical data in Figure 5a.

For example, the expected availability time of node $v$ is six hours, we can get the probability that node $v$ will keep the state of donating resources for at least six hours is 50% according to Figure 5a and the node $v$ will keep the state of donating resources for at least three hours is 100%. Similarly, we can calculate the different probabilities corresponding to different actual availability time of the node $v$. Therefore, we can obtain the fuzzy membership function image of the actual availability time of the node $v$ according to Figure 5a, and the fuzzy membership function image of node $v$ whose expected availability time is six hours is shown in Figure 5b. We can calculate the lower bound of "Range" is 3 according to Figure 5b. Therefore, we abstracted out two parameters as mentioned in Section 3.1 to describe node $v$ and calculated the confidence interval by their values.

| $ratio_v$ | $probability_v$ |
|-----------|-----------------|
| <=50% | 100% |
| 60% | 90% |
| 70% | 80% |
| 80% | 70% |
| 90% | 60% |
| 100% | 50% |

(**a**)

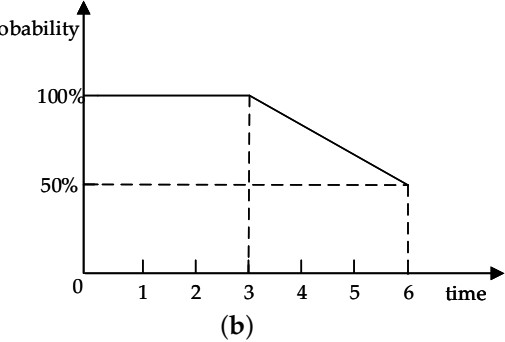

(**b**)

**Figure 5.** The statistics of actual availability time and expected availability time of the node $v$ and membership function image of node $v$ expected availability time which is six hours. (**a**) Statistics of actual availability time and expected availability time of the node $v$; (**b**) The membership function image of the node $v$ expected availability time which is six hours.

By analyzing the ratio of the actual availability time of multiple nodes to their expected availability time, we found that the parameter $v_j.trust$ of the node decreases monotonically with the parameter $h_j$. For simplicity, suppose that the parameter $v_j.trust$ is linearly decreasing with the parameter $h_j$, and the decreasing function is denoted by $\Phi(t)$. However, the nodes' behaviors are uncertain in VCPs, so the decreasing function can be designed in the form of a nonlinear decreasing function, as mentioned in the literature [13,21]. Although the function image of each VN is different because of its historical performance, for easy calculation, we assume that the decreasing function satisfies the equation as follows:

$$\Phi(t + \Delta t) - \Phi(t) = -0.5\Delta t \tag{1}$$

For example, the parameters of the node $v_2$ are shown in Figure 3b, $v_2.trust = 0.3, h_j = 4$, and the decreasing function image of the node $v_2$ is shown in Figure 6a. Suppose that the current time is 0, the confidence interval of the node $v_2$ is $Range_2[2.6, 4]$ according to its decreasing function. From the confidence interval of the node $v_2$, it can be seen that it will keep the state of donating resources within 2.6 h, and it will disconnect at any time in the time interval from 2.6 to 4. Similarly, it can be applied to the confidence interval of the other nodes in Figure 3b, as shown in Figure 6b.

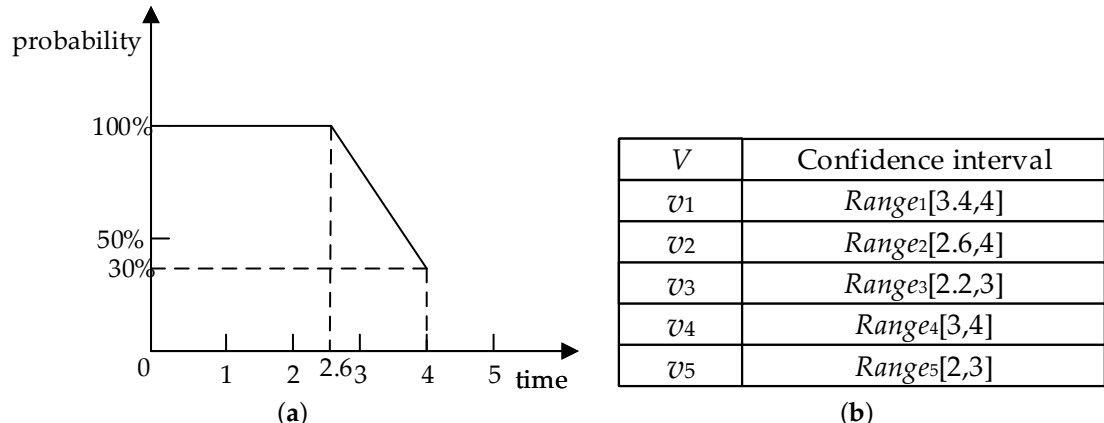

| $V$ | Confidence interval |
|---|---|
| $v_1$ | $Range_1[3.4,4]$ |
| $v_2$ | $Range_2[2.6,4]$ |
| $v_3$ | $Range_3[2.2,3]$ |
| $v_4$ | $Range_4[3,4]$ |
| $v_5$ | $Range_5[2,3]$ |

(a)          (b)

**Figure 6.** The decreasing function image of the node $v_2$ and the confidence interval of each node. (**a**) The decreasing function image of the node $v_2$; (**b**) The confidence interval of each node.

The objective of our task scheduling is to minimize the total completion time under the available resource constraints. The available resource constraints are as follows:

$$D(v_j) <= h_j, \tag{2}$$

where $D(v_i)$ represents the total time assigned to the node $v_j$. Constraint (2) ensures that the total time assigned to the node $v_j$ must not exceed the expected time provided by the node $v_j$.

In this paper, we use the reliability model to reduce the gap between the expected total completion time and the actual total completion time, which makes the expected minimum total completion time approximated to the actual minimum total completion time.

## 4. Algorithm Description

In this section, we introduce the EFTT and IEFTT algorithms in detail.

*4.1. The EFTT (Earliest Finish Task Based on Trust) Algorithm*

The EFTT algorithm is a task scheduling algorithm under the constraints of available resources and prerequisite task sets, it minimizes the makespan and has two phases: a *calculation phase of task execution order* for computing the order of assignment and a *node selection phase* for selecting the most suitable VNs according to task priority.

**The calculation phase of the task execution order:** This phase calculates the task priority of each task according to the DAG. In this paper, task priority is equal to the distance between a node and the root node in the DAG. Then, this phase sorts the tasks in descending order according to task priority, and the sort result is the task execution order R. If the tasks have the same task priority, we sort the tasks according to their order of joining (i.e., first-come-first-scheduled policy).

Significantly, in this phase, the task is assigned according to task priority in the task set T. When task $t_i$ is assigned, all prerequisite tasks need to be completed. Otherwise the task cannot be assigned and the tasks that need to be redone will be assigned first in the next round.

**The node selection phase:In the node selection phase**, the EFTT algorithm obeys the following two principles:

- **Principle 1 (confidence interval preference principle)** since the volunteer node is often disconnected, which will make a turnaround time of a task become slower. To reduce its impact, the EFTT algorithm selects the confidence interval first to assign the task.
- **Principle 2 (task allocation principle)** to ensure that the prerequisite task $t_i$ has been completed when task $t_j$ starts to be executed, the EFTT algorithm first schedules all the computing resources to calculate task $t_i$. In this way, task $t_i$ will be divided into $|V|$ equal parts, $|V|$ is the number of online volunteer nodes, and each volunteer node is assigned a time interval to calculate task $t_i$ The size of each time interval is $t_i.cost/|V|$.

Suppose the current time m is 0. Firstly, the EFFT algorithm calculates the confidence interval of each node according to the descending function of each volunteer node. Secondly, the EFFT algorithm determines the computation time $l'$ of task $t$ according to the cost of $t$ and the number of volunteer nodes $|V|$.

For a volunteer node $v$ in the volunteer node set $V$, if the time interval assigned to task $t$ is within $v's$ confidence interval, that is, $m + l' < l_a$, we could believe that task $t$ assigned to a volunteer node $v$ can be completed; if the time interval assigned to task $t$ exceeded $v'$ s confidence interval, that is, $m + l' > l_a$, according to confidence interval preference principle, volunteer node $v$ will stop computing task $t$ at time $l_a$ and it will be added to non-confidence node set $V'$.

If $V$ was null at time $l_1$, we could believe that the confidence interval of VNs have been completely allocated. If there was still uncompleted task $t'$, $t'$ would be assigned to the node $v'$ with the largest trust value in $V'$ until the node $v'$ disconnect or the task is completed. Specifically, the EFTT algorithm is described in detail in Algorithm 1. To select the most suitable node to assign tasks, we use bubble sort algorithm to preprocess data. In the future, considering the time complexity of our algorithms, other sorting algorithms such as quicksort and parallel sorting algorithm [35] will be adopted to use in pre-processing data.

To facilitate a clearer understanding of the implementation process of Algorithm 1, examples are as follows:

---

**Algorithm 1** The EFTT algorithm

---

**Input:** volunteer node set $V$, task set $T$ and corresponding TAG, the current time $m = 0$
**Output:** task assignment set $T.assign$
 1: Calculate the task execution set $R$
 2: $V' = \varnothing$
 3: **for each** $v \in V$ **do**
 4:      Calculate the confidence interval of $v$
 5: **end for**
 6: **while** $R \neq \varnothing$ and $(V' \neq \varnothing$ or $V \neq \varnothing)$ **do**
 7:      Take the first task $t'$ form $T$
 8:      **if** $V \neq \varnothing$ **then**
 9:          $l' = t'.cost / |V|$
10:          **for** each $v \in V$ **do**
11:             **if** $l_a > m + l'$ **then**
12:               Add $< v, l' > toT.assign$
13:               $t'.cost = t; .cost - 1$
14:             **else**
15:               Add $< v, l_a - m > toT.assign$
16:               Add $v$ to $V'$
17:               $t'.cost = t'.cost - (l_a - m)$
18:             **end if**
19:          **end for**
20:          $m = m + l'$
21:      **else**
22:          **while** $V' \neq \varnothing$ **do**
23:          Take $v'$ who has the biggest $v.trust$ in $V'$//use bubble sort algorithm to find the maximum value
24:          **if** $v'$ is offline **then**
25:             **Continue**
26:          **else**
27:             $t'.cost = t'.cost - (l_b - m)$
28:             $m = m + (l_b - m)$
29:             Add $< v', l_b - m > 0$ to $T.assign$
30:          **end if**
31:          **if** $t'.cost = 0$ **then**
32:             Delete $t'$ from $R$
33:          **end if**
34:          **end while**
35:      **end if**
36: **end while**
37: **return** $T.assign$

---

    Given a task set $T$, a volunteer node set $V$ and the DAG are as shown in Figures 2 and 3. According to task priority, we can calculate task execution order set $R = \{t_1, t_2, t_3, t_4, t_5, t_6\}$. Suppose the current time $m = 0$. In the EFTT algorithm, **step one:** the task $t_1$ is assigned, since $t_1.cost / 5 = 0.4$. Since all the nodes satisfy the inequality $m + l' < l_a$, all nodes can complete $t_1$ under the constraints of their confidence intervals. After $t_1$ is completed, $m$ is updated to 0.4. **Step two:** it can be seen that all nodes can complete tasks $t_2$ and $t_3$ under their confidence interval constraints. After $t_3$ is completed, $m$ is updated to 1.2. **Step three:** the EFTT algorithm takes task $t_4$ from $R$, since $t_4.cost = 4, l' = t_4.cost / 5 = 0.8$, and $l_a$ of node $v_5$ is not greater

than $m + l' = 1.2 + 0.8 = 2$, the EFTT algorithm deletes $v_5$ from the node set $V$ and add it to the set $V'$. At this time, $m$ is updated to 2. **Step four:** the EFTT algorithm takes task $t_5$ from $R$, since $t_5.cost = 2, l' = t_5.cost/4 = 0.5$, and $l_a$ of the node $v_3$ is not greater than $m + l' = 2 + 0.5 = 2.5$. At the same time, departure of the node $v_3$ at time 2.2, so $v_3$ provides the calculation time is 0.2, and the task $t_5$ cannot be completed at time 2.5. **Step five:** the EFTT algorithm takes the rest of the task $t_5$ to assign, $t_5.cost = 2 - 1.7 = 0.3, l' = t_5.cost/3 = 0.1$, since the $l_a$ of the node $v_2$ is not greater than $m + l' = 2.5 + 0.1 = 2.6$, the EFTT algorithm deletes $v_2$ from the node set $V$ and add it to the set $V'$. At this time, $m$ s updated to 2.6. **Step six:** the EFTT algorithm takes task $t_6$ from $R$, since $t_6.cost = 3, l' = t_6.cost/2 = 1.5$, and $l_a$ of the node $v_1$ and node $V_4$ are not greater than $m + l' = 2.6 + 1.5 = 4.1$, the EFTT algorithm deletes $v_1$ and $v_4$ from the node set $V$ and add them to the set $V'$. At this time, $m$ is updated to 2.6. **Finally,** the EFTT algorithm continually selects the node in $V'$ to calculate the rest of $t_6$ until the node $v'$ is empty and then the algorithm stops. The specific task assignment is shown in Figure 7.

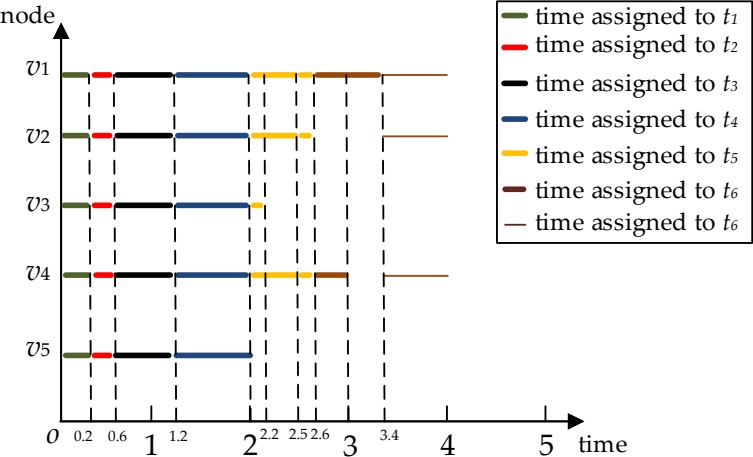

**Figure 7.** The specific task assignment of the EFTT algorithm.

*4.2. The IEFTT (Improved Earliest Finish Task Based on Trust) Algorithm*

The basic scheduling algorithm EFTT establishes a reliability model by associating a trust value with each volunteer node to improve the performance of task assignment and minimize the makespan. However, the EFFT algorithm results in some idle computing resources and a larger number of task fragments. On the basis of the EFFT algorithm, this paper proposes an improved scheduling algorithm IEFTT. The IEFTT algorithm optimizes task scheduling in two ways: task completion time and task fragment rate. Although our second algorithm, the IEFTT algorithm shown in Algorithm 2, has a *calculation phase of task execution order and a node selection phase,* it uses a different method for selecting the most suitable VN for each selected task.

To reduce task segmentation rate, the concept of time period is proposed and the IEFTT algorithm divides variable-sized tasks into small-sized tasks according to task priority. **Firstly,** according to the confidence interval of each VNs, the IEFTT algorithm divides up total computing time donated by VNs into small-sized time periods. **Secondly,** the IEFTT algorithm adds the left and right endpoints of the nodes' confidence intervals to the set $L$ in ascending order. After removing all duplicate elements from the set $L$, the IEFTT algorithm takes every two adjacent elements of the set $L$ to form a time period. **Thirdly,** the IEFTT algorithm calculates total computing time of each time period, and the total computing time of time period $n$ is denoted by $V_n.con$. **Fourth,** the IEFTT algorithm takes out the tasks with the highest task priority from the task execution order set $R$. Suppose the highest task priority denoted by $p$ and total computing time of the tasks with task priority $p$ is $T_{(p)}.cost$. **Finally,** the IEFTT algorithm compares the value between $V_n.con$ and $T_{(p)}.cost$, if $V_n.con$ is larger $T_{(p)}.cost$, then the value of task slice is $T_{(p)}.cost/|V|$, otherwise the value of task slice is $V_n.con/|V|$. Moreover, the IEFTT

algorithm did not stop until that the task execution order set $R$ was empty or the volunteer set V was empty. The IEFTT algorithm is described in detail in Algorithm 2.

---

**Algorithm 2** The IEFTT algorithm

---

**Input:** task set $T$ and its corresponding TAG at time $l_1$, the volunteer set $V$ the number set $V$ is $|V|$
**Output:** task assignment $T.assign$.

1: Calculate the task execution set $R$
2: $L = \varnothing, T' = \varnothing, T'_u.cost = 0, u = 0$
3: **for each** $v \in V$ **do**

4:    Calculate the confidence interval $Range[l_a, l_b]$ of $v$
5:    *Add $l_a$ and $l_b$ to L*
6: **end for**
7: Remove all duplicate elements from the set $L$ and take every two adjacent elements of the set $L$

    to form a time period
8: the number of time period is $|L|$ and the total computing time of time period $n$ is $V_n.con$
9: **for each** the $nth$ time period $\in$ total $|L|$ time period **do**

10:    Calculate the total computing time of time period $n$ is $V_n.con$
11:    **while** $V_n.con > 0$ **do**

12:        Call the Algorithm 3 ***ComTSet***$(V_n.con, R, |V|, T')$ // calculate the size of task slice $v.compute$
13:        **for each** $v \in V$ **do**

14:            **for** each $t \in R$ **do**

15:                $u = u + 1$
16:                $T'(u).cost = T'(u).cost + tu.cost$ // *$T'(u).cost$ represents the total computing time of the tasks*

                *from the 1st task to the uth task in task execution order set R.*
17:                **if** $T'(u).cost >= v.compute$ **then**

18:                    *break*
19:                **else**

20:                    Add $< tu, v >$ to$T.assign$
21:                **end if**
22:            **end for**
23:            **if** $T'(u).cost >= v.compute$ **then**

24:                The task $t_u$ continues to be divided up into two task slices, whose sizes are

                $T'(u).cost$-$v.compute$ and $t_u.cost$-$(T'(u).cost$-$v.compute)$
25:            **end if**
26:            **if** $v$ keeps online **then**

27:                Delete the first $(u - 1)$ task from the set $T'$
28:            **else**

29:                Delete $v$ from the set $V$
30:            **end if**
31:        **end for**
32:    **end while**
33: **end for**

---

To facilitate a clearer understanding of the implementation process of Algorithm 2, examples are shown as follows:

Given a task set $T$, a volunteer node set $V$, the DAG and the confidence interval of each node are as shown in Figures 2, 3 and 6. According to task priority, we can calculate task execution order set $R = \{t_1, t_2, t_3, t_4, t_5, t_6\}$. Suppose the current time $m = 0$. In the IEFTT algorithm, **Step one:** According to line 7 of the Algorithm 2, the set $L$ is $\{2, 2.2, 2.6, 3, 3.4, 4\}$, and the set $L$ forms six time periods, which are [0,2], [2,2.2], [2.2,2.6], [2.6,3], [3,3.4] and [3.4,4] as are shown in Figure 8. **Step two:** the IEFTT

algorithm calculates the total computing time of the first time period $V_1.con = 10$, and the total computing time of the tasks with task priority 1 is $T_{(1)}.cost = 6$. **Step three:** the IEFTT algorithm compares the value of $V_1.con$ and $T_{(1)}.cost$, and according to the Algorithm 3, the task slice $v.compute$ is 6/5 in the first time period and the current task set to be executed is $T' = \{t_1, t_2, t_3\}$. **Step four:** According to lines 21 to 23 of the Algorithm 2, the task $t_1$ continues to be divided up into two different task slices, whose values are 1.2 and 0.8, and the two task slices are assigned to the volunteer node $v_1$ and $v_2$. Similarly, the assignment of task $t_2$ and $t_3$ can be calculated, as shown in Figure 9. **Step five:** after completing task $t_1, t_2$ and $t_3$, $V_1.con = 4$, the IEFTT algorithm takes out tasks whose task priority is 2 to assign according to lines 11 and 12 of the Algorithm 3. Since $T_{(2)}.cost = 6$ and $V_1.con < T_{(2)}.cost$, the IEFTT algorithm compares the values between $V_1.con$ and $T_{(1)}.cost$ and according to the Algorithm 3, the task slice $v.compute$ is 4/5 in the left of its first time period and the current task set is to be executed $T' = \{t_4, t_5\}$. **Step six:** According to line 12 of the Algorithm 3, the task t4 continues to be divided up into five task slices, whose values are 0.8, and the five task slices are assigned to the volunteer node $v_1, v_2, v_3, v_4$ and $v_5$ correspondingly. **Finally,** after completing the task $t_1, t_2$ and $t_3$, $V_1.con = 0$, the IEFTT algorithm did not stop assigning the left tasks in the next time period in a similar way until $V_1.con$ was zero. The specific task assignment is shown in Figure 9.

---

**Algorithm 3** The task slice function *ComTSet (Vn.con,R,|V|,T')*

---

**Input:** the total computing time of **time period** $n$ is *Vn.con, task execution order set R*, the number
　　of the volunteer nodes, the current task set $T'$ to be executed
**Output:** the size of task slice $v.compute$,
　1: Seek the highest task priority $p$ from $R$
　2: $T'.cost = 0$ //$T'.cost$ represents the total computing time of the current task set $T'$ to be executed
　3: **while** $T'.cost < Vn.con$ **and** there is also the task with task priority $p$ in $R$ **do**
　4:　　Take the task $t'$ with task priority $p$ from $R$
　5:　　Add $t'$ the set $T'$
　6:　　$T'.cost = T'.cost + t'.cost$
　7: **end while**
　8: **if** $Vn.con < T'.cost$ **then**
　9:　　$v.compute = V_n.con/|V|$
　10: **else**
　11:　　$v.compute = T'.cost/|V|$
　12: **end if**
　13: **return** $v.compute$

---

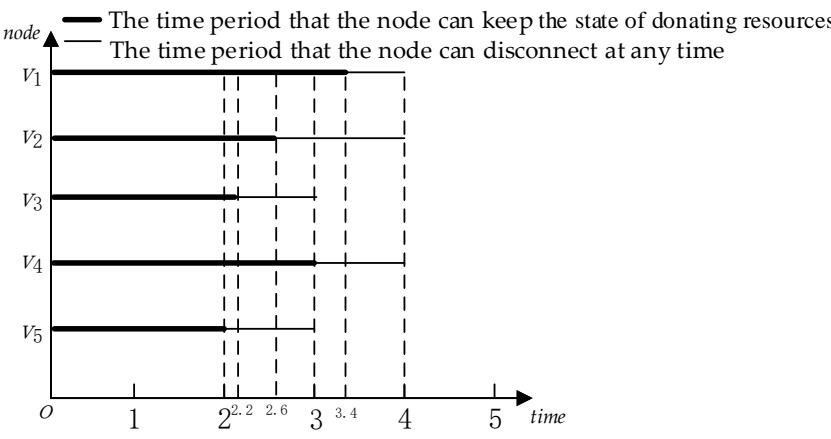

**Figure 8.** Time periods of the set *L*.

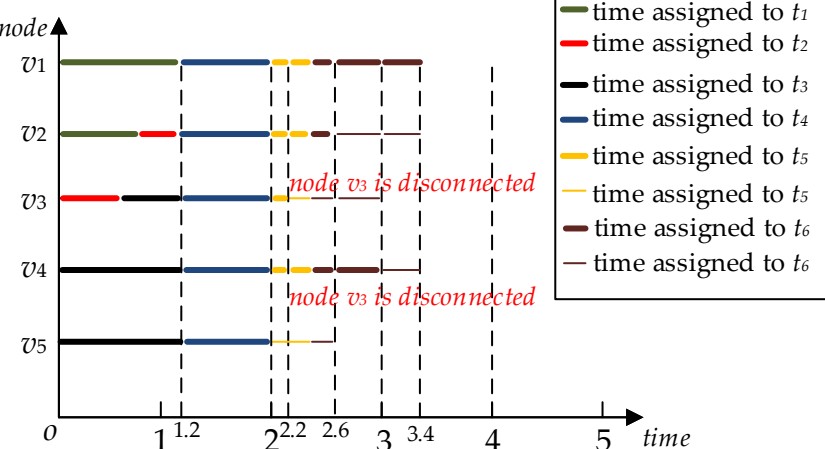

**Figure 9.** The specific task assignment of the EFTT algorithm.

## 5. Experimental Evaluation

In this section, we use a static task set and dynamic task set to test the performance of the proposed algorithms and compare the results with the HEFT-AC algorithm and the HEFT-ACU algorithm mentioned before. This paper conducts experiments to simulate the volunteer computing environment. In the simulation experiment environment, we used one master node and fifty sub nodes. All nodes are configured with Intel Core i7 4790 CPU@3.4GHZ, 8GB DDR3 memory, 1TB hard disk and Windows 10 operating system. To be closer to the real-world volunteer computing environment, 15–30 threads are opened on each host to simulate the nodes and the number of this type of nodes will be between 750–1500. At the same time, we need to calculate a membership function for each VN. Specifically, we adopted the interpolation method to get the trust decreasing function.

### 5.1. Experimental Results and Analysis on Static Task Sets

In the experiment with a static task set, three common tasks were used: word frequency statistics, inverted index and distributed Grep. The input files are the data and dump files provided by Wikipedia (the main contents are entries, templates, picture descriptions and basic meta-pages, etc.). We mainly consider the influence of two main parameters as follows:

- The task set scale which is the number of tasks included in the task set $T$.
- The average size of tasks in task set $T$ measured by the number of task input file fragments.

We assume that the size of a task fragment is 64 MB and the completion time of each task fragment is 70 s. Table 2 shows the default values and ranges of the main parameters.

**Table 2.** Experimental default parameters.

| Parameter | Default | Value Range |
|---|---|---|
| average size of tasks(MB) | 128 | 64–320 |
| task set scale | 500 | 200–1000 |
| number of VN | 1000 | 750–1500 |

In this paper, the task completion time of set $T$ is the primary performance index. In addition, this paper also uses the rework rate to measure the performance of the algorithm more comprehensively. Obviously, the lower the rework rate is, the higher the system reliability is. The rework rate is defined as follows:

$$\text{rework rate} = \text{the number of rework tasks} / \text{the number of tasks in set} T, \qquad (3)$$

To evaluate our algorithms, we choose HEFT-AC and HEFT-ACU as comparison algorithms. This is because these two algorithms are the most relevant to our study. At the same time, these two algorithms are more effective than many famous algorithms, so we choose these two algorithms for comparison. In addition, in this paper, the objective of our work is to reduce the rework rate and complete all tasks as soon as possible. Therefore, to evaluate the performance of our algorithms, we take task completion time and rework rate as performance indicators and compare them with HEFT-AC and HEFT-ACU. The two algorithms are described briefly below.

The HEFT-AC algorithm calculates the completion time of the first unscheduled task completed by each VN at each step, and it selects the VN with the minimum completion time to assign. This process does not stop until all the tasks are scheduled. On the basis of the HEFT-AC algorithm, the HEFT-ACU algorithm selects the VN with the maximum reputation value to assign the first unscheduled task of a set. The HEFT-ACU algorithm can reduce the rework rate. Since the two algorithms mainly focus on the independent task assignment, we adjust the two algorithms to deal with the assignment of the task priority so that we can compare the results with our proposed algorithms.

### 5.1.1. The Impact of Average Size of Tasks

As shown in Figure 10, we test the impact of the different average size of tasks on the performance of the four algorithms. It can be seen that IEFTT performs the best among the four algorithms in both total completion time and rework rate, and the HEFT-ACU algorithm is slightly worse than the IEFTT algorithm. The other two algorithms are much less efficient. This is because the IEFTT algorithm and the HEFT-ACU algorithm take the system reliability into account. Even if they encounter a suddenly offline volunteer node, they can still give a flexible task assignment.

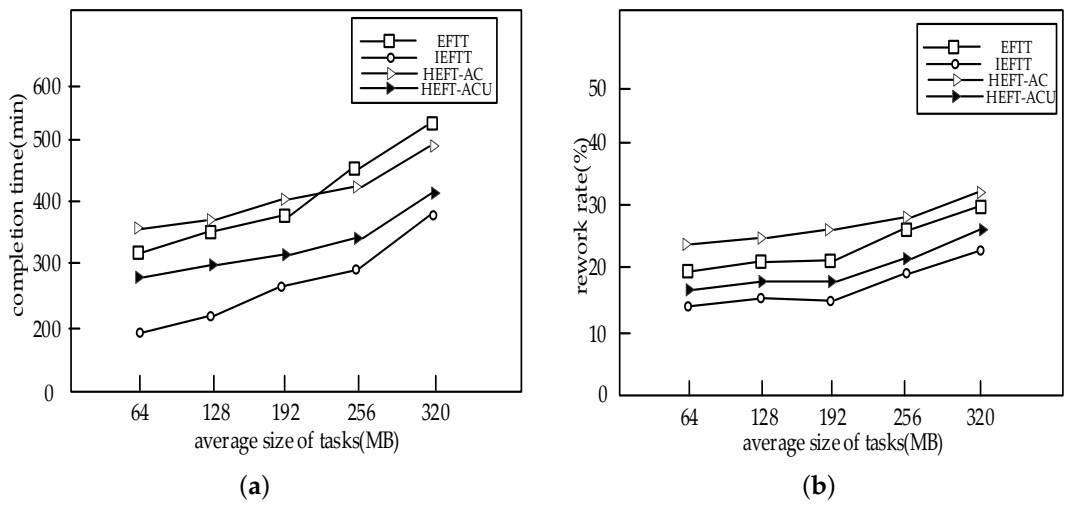

**Figure 10.** The impact of average size of tasks on the performance of the algorithms: (**a**) The impact of average size of tasks on task completion time; (**b**) The impact of average size of tasks on rework rate.

### 5.1.2. The Impact of Task Set Scale

Figure 11 tests the effect of the task set scale on the performance of the algorithms. With the increase of task set scale, the completion time will gradually increase. However, the rework rate does not change significantly when the task set scale reaches a certain size. This also fully demonstrates the robustness of the proposed algorithms. As expected, the IEFTT algorithm has the best performance among the four algorithms.

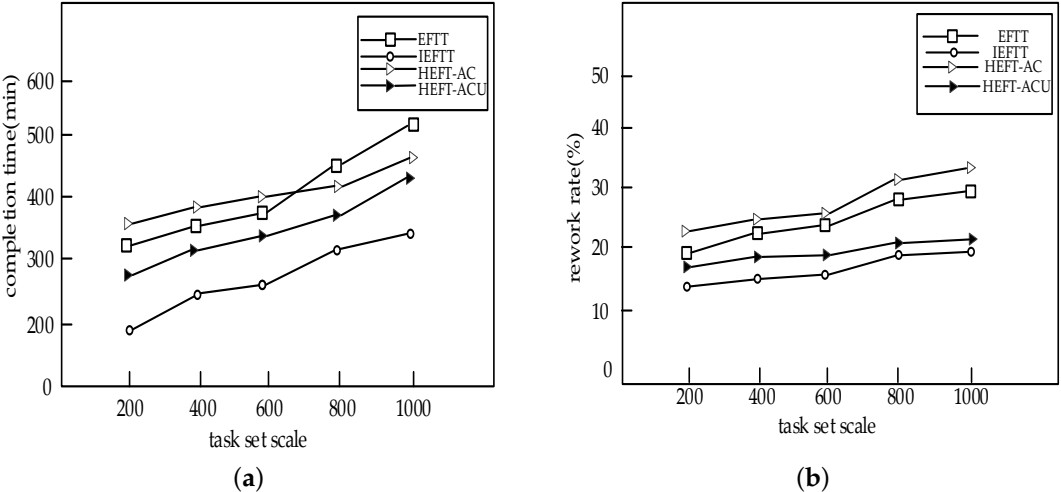

**Figure 11.** The impact of task set scale on the performance of the algorithms: (**a**) The impact of task set scale on task completion time; (**b**) The impact of task set scale on rework rate.

### 5.1.3. The Impact of the Number of Volunteer Nodes

In Figure 12, we analyze the impact of volunteer nodes. It can be noticed that the number of volunteer nodes increase, while both the completion time and the redo rate decrease. This is because the more volunteer nodes exist in a volunteer platform, the more computing power it has in a volunteer platform, and more computing resources can be selected.

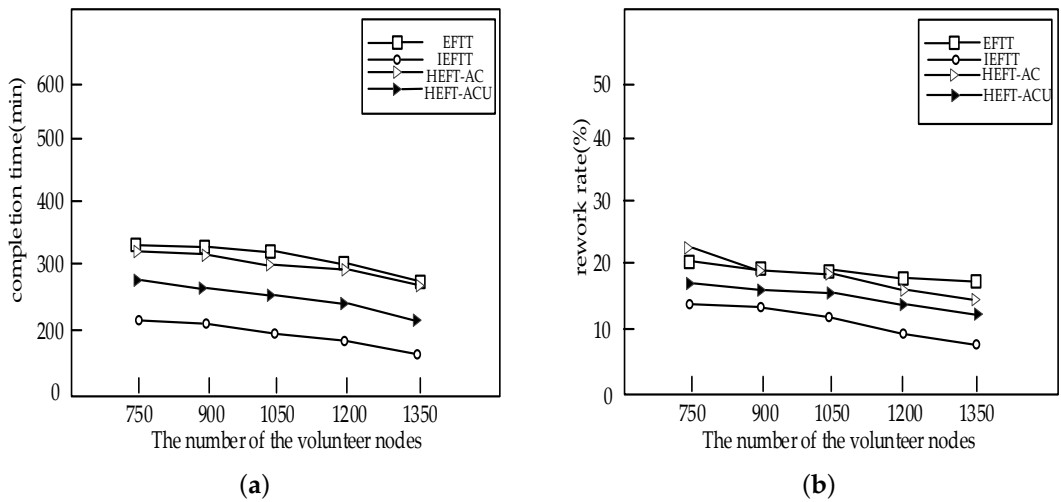

**Figure 12.** The impact of the number of volunteer nodes on the performance of the algorithms: (**a**) The impact of the number of volunteer nodes on completion time; (**b**) The impact of the number of volunteer nodes on rework rate.

### 5.2. Experimental Results and Analysis on Dynamic Task Sets

In order to be closer to the real application scenario, this section uses a dynamic set of application tasks. The experiment generated forty task sets, and the size of each task fragment is 64 MB. In the experiment, the system is monitored every 10 min to obtain the number of tasks completed and task rework rate.

Figure 13 shows the experimental results. It can be seen that the IEFTT algorithm has obvious advantages on dynamic task sets, regardless of the number of tasks completed or the task rework rate. In particular, the task rework rate of the EFFT algorithm is lower than that of IEFFT algorithm,

this is mainly because at the beginning EFFT algorithm dose not select the interval that the volunteer node can be offline suddenly. Through the above experimental results, the validity of the algorithms proposed in this paper is further proved.

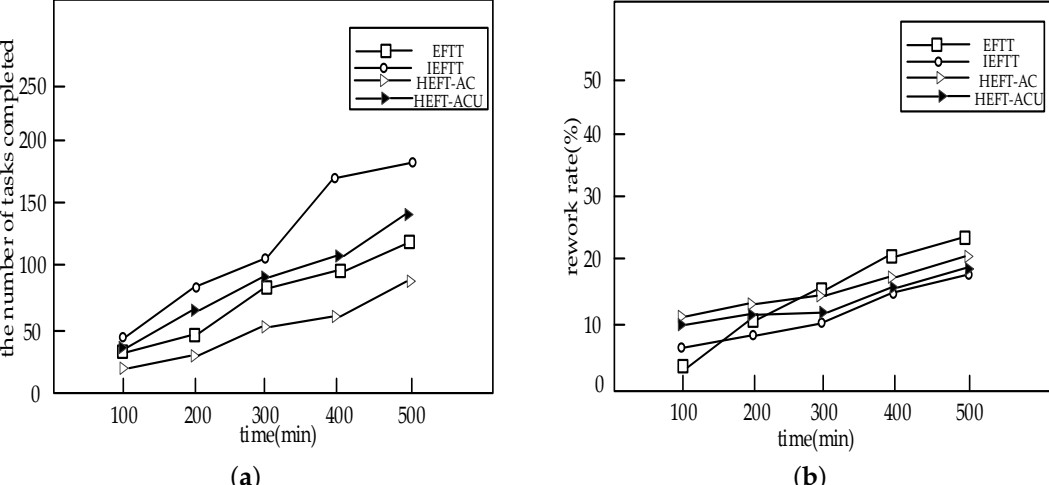

**Figure 13.** Performance comparisons of the algorithms on dynamic task set: (**a**) Comparison of the number of task completed; (**b**) Comparison of the rework rate.

## 6. Conclusions

In this paper, to complete all tasks as soon as possible by considering system reliability, firstly, we proposed a reliability model, which can calculate the time interval that volunteer node can keep the state of donating resources. Secondly, on this basis, we propose an EFTT algorithm that satisfies the task priority constraint to complete all the tasks as soon as possible. However, the EFFT algorithm results in the idling of some computing resources and a high task segmentation rate. Thus, on the basis of the EFFT algorithm, we propose an improved scheduling algorithm IEFTT. Finally, compared with the well-known algorithms, our proposed algorithms can complete the task faster. Moreover, the study of efficient task assignment in volunteer computing has important practical significance, which not only provides convenient conditions for the analysis and processing of big data, but can also be applied to high-performance computing in a small range, such as the management of computing resources in university laboratories. In future, we will consider more factors that may affect task scheduling in VCPs such as heterogeneity, dynamics and scalability.

**Author Contributions:** L.X. designed and wrote the paper; J.Q. supervised the work; L.X. performed the experiments; S.L. and R.Q. analyzed the data. All authors have read and approved the final manuscript.

**Funding:** This work was supported by the National Social Science Foundation of China (No. 15BYY028) and Dalian University of Foreign Languages Research Foundation (No. 2015XJQN05).

**Conflicts of Interest:** The authors declares no conflict of interest.

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
