# Peer review of "Task Assignment Algorithm Based on Trust in Volunteer Computing Platforms"

_information, doi:10.3390/info10070244_

Round 1

Reviewer 1 Report

I'm pleased to see research on volunteer computing,

and I appreciate that considerable work went into this paper.

However, the work is based on invalid assumptions,

and for this reason I don't think that the results are of practical value,

at least in the context of volunteer computing.

The central problem is the assumption that if a volunteer node

"suddenly goes offline" - as the paper puts it - any jobs it's running fail.

That's not the case.

In BOINC, computing on a volunteer node can stop for a number of reasons:

the computer is in use, or the non-BOINC CPU load exceeds a threshold,

or a user-selected program such as a game is running.

In all these cases, running jobs are suspended,

and are resumed when the cause of the stoppage goes away.

The period of stoppage maybe be a few minutes, or even a few seconds.

Even if the volunteer node is powered off, running jobs don't fail.

Most BOINC applications checkpoint every few minutes;

for those based on virtual machine technology,

this happens automatically using VM "snapshots".

In any case, then the node is powered on again,

the jobs resume more or less where they left off.

So the net effect of "going offline" is not that jobs fail,

but that they run slower - perhaps by only a small factor.

BOINC maintains statistics of this effect,

and uses it to estimate job completion.

The paper assumes that workloads are DAGs,

and that the goal is to finish each DAG as fast as possible.

In practice, however, volunteer computing is used entirely for high-throughput computing:

i.e. the goal is to maximize the rate of job completion,

not to minimize the turnaround time of a job or group of jobs.

By the way, some work has been done in minimizing the makespan

of batches of jobs in the context of volunteer computing,

i.e. dealing with the "straggler problem".

I don't see this in the references.

The paper's 2nd algorithm assumes that tasks are arbitrarily divisible.

This is not the case for most or all existing applications of volunteer computing.

In addition, the paper has some problems in its content and presentation.

It uses a number of terms without precisely defining their meaning,

such as "reliability", "system reliability", "trust", "system redundancy mechanism",

"reputation", and others.

Similarly, it refers to "problems" without saying exactly what the problem is.

I think I got the general idea of how the 2 algorithms work,

but I was unable to follow the details, and the pseudocode is not that helpful.

In cases like this, it's best to give a clear, intuitive idea of

how and why the algorithm works before giving details or code.

In summary: I can't recommend the paper for publication in its current state.

If the presentation were improved

(define terms before using them, state problem clearly, motivate algorithms better)

then I wouldn't object to publication,

though as I said its relevance to actual volunteer computing is dubious.

I certainly don't want to discourage the authors from doing research in volunteer computing.

My longer-term recommendation is that they become involved in

the BOINC community of scientists and developers,

and identify research problems based on actual practice;

there are many such problems.

Author Response

Response to Reviewer 1 Comments

The authors have very carefully followed all constructive comments from the managing editor and two reviewers, and have tried every possible effort to clean up every issue that was raised by them. We provide below a detailed account on the changes that we have made in response to every comment that the reviewers have provided.

The authors would like to thank the managing editor and two reviewers for their constructive comments on this manuscript and positive support.

For all corrections, we highlight them in the new manuscript.

Q1. I'm pleased to see research on volunteer computing, and I appreciate that considerable work went into this paper. However, the work is based on invalid assumptions, and for this reason I don't think that the results are of practical value, at least in the context of volunteer computing.

The central problem is the assumption that if a volunteer node "suddenly goes offline" - as the paper puts it - any jobs it's running fail. That's not the case.

In BOINC, computing on a volunteer node can stop for a number of reasons: the computer is in use, or the non-BOINC CPU load exceeds a threshold, or a user-selected program such as a game is running. In all these cases, running jobs are suspended, and are resumed when the cause of the stoppage goes away. The period of stoppage maybe be a few minutes, or even a few seconds.

Even if the volunteer node is powered off, running jobs don't fail. Most BOINC applications checkpoint every few minutes; for those based on virtual machine technology, this happens automatically using VM "snapshots". In any case, then the node is powered on again, the jobs resume more or less where they left off.

So the net effect of "going offline" is not that jobs fail, but that they run slower - perhaps by only a small factor. BOINC maintains statistics of this effect, and uses it to estimate job completion.

Authors’ reply:

Thank you very much for your advice. Yes, our assumption is not the case in normal volunteer computing. Moreover, computing on a volunteer node can stop for a number of reasons as you mentioned and the running jobs don't fail. We have reviewed our manuscript carefully and rechecked a large number of papers about volunteer computing according to your suggestions. We found the motivation of our proposed algorithm in our old manuscript is problematic. Because volunteer computing is composed of donated resources, they can make few guarantees about network or machine reliability. Therefore, the expected completion time is often disturbed, and a turnaround time of tasks is usually days or months. In this paper, we propose algorithms for minimizing a turnaround time of batches of tasks and reducing the disturbance by using a flexible task assignment, rather than used in a normal volunteer computing.For minimizing a turnaround time of batches of tasks in a volunteer computing platform, the key is ensuring that all tasks are distributed to suitable workers in a timely manner and that workers complete the tasks as soon as possible. In normal volunteer computing, where workers request tasks from the master, which is not applied to our algorithms. In our algorithms, the master can assign tasks to arbitrary workers. Because a worker machine may often be disconnected from the network, used for other purposes or completely quit the computation without advanced warning, we associate a trust value with each volunteer node similar to the HEFT-ACU algorithm [34] to reduce its disturbance of makespan.Some early volunteer computing applications such as SETI@home attracted about on the order of 1M volunteers. However, volunteers have shrunk to 200K or so because of little media coverage or other reasons. Therefore, volunteer computing faces the challenge of marketing itself to the broader public. Some possible approaches such as partnerships with other distributed computing to meet their special requirement. Although our algorithms are not applied to normal volunteer computing, they may be useful for the extension of volunteer computing. In the future, we will be actively involved in the BOINC community of scientists and developers, and identify research problems based on actual practice. For clarity, we have modified the motivation of our proposed algorithms and rewritten the abstract and introduction. The modified paragraphs are shown as follows: 
Abstract: In volunteer computing (VC), the expected availability time and the actual availability time provided by volunteer nodes (VNs) are usually inconsistent. Scheduling tasks with precedence constraints in VC under the situation is a new challenge. In this paper, we propose two novel task assignment algorithms to minimize completion time (makespan) by a flexible task assignment. Firstly, this paper proposes a reliability model, which uses a simple fuzzy model to predict the time interval provided by VN. This reliability model can reduce inconsistencies between the expected availability time and actual availability time. Secondly, based on the reliability model, this paper proposes an algorithm called EFTT (Earliest Finish Task based on Trust, EFTT), which can minimize makespan. However, EFTT may induce resource waste in task assignment. To make full use of computing resources and reduce task segmentation rate, an algorithm IEFTT (improved earliest finish task based on trust, IEFTT) is further proposed. Finally, experimental results verify the efficiency of the proposed algorithms. 
The 2nd paragraph in Section 1“With VC is recognized for high-throughput computing, more and more scientific problems were deployed in VCPs. Because donated resources of VN are not available all the time, the completion time (makspan) of a task is usually days or months. Therefore, the goal of many task assignment algorithms in prior studies [6-9] was to maximize the rate of task completion. However, as mentioned in [10,11], minimizing the turnaround time of a task or batches of tasks was very meaning for some special VC applications. Moreover, VC Applications such as SETI@home should be more efficient if they use minimizing a turnaround as the goal of task assignments. However, there are few studies about makespan in VC and the prior studies didn’t pay much attention to the reliability in task assignment.” 
The 3rd paragraph in Section 1“Reliability is often associated with fault tolerance, and fault tolerance is often associated with task failure or malfunctioning workers. Task failure or malfunctioning workers can be solved by redundant tasks or other techniques such as rescheduling [12]. Although these techniques are helpful for reliable task assignment, the performance of scheduling would be improved if reliability is taken into account in task assignment.” 
The 4th paragraph in Section 1“Usually, reliability of the distributed systems is defined as the probability that task can be executed successfully [13]. In the current context for task assignment, the definition can be narrowed down to the gap between the expected availability time and actual availability time. The gap characterizes the performance of task assignment, and the smaller the gap, the better the performance of task assignment. Because VC differs from other distributed computing such as grid computing and cloud computing. In VC, the expected availability time of VN is often disturbed by some reasons, such as the non-VC CPU load exceeds a threshold or other purpose. In such circumstances, there exists the gap between the actual makespan and the expected makespan, which causes the performance degradation of task assignment. Consequently, reliability is very useful for task assignment of VCPs.” 
The 5th paragraph in Section 1

In addition, many prior studies [7,9,14] about task assignment focused on independent tasks. However, tasks of some VC applications in practice are precedence-constrained.

The 6th paragraph in Section 1“In this paper, we address the task assignment problem in VCPs and propose two task assignment algorithms that take into account reliability, makespan, and tasks with precedence-constrained. The main contributions of this paper are summarized as follows:”

Q2. The paper assumes that workloads are DAGs, and that the goal is to finish each DAG as fast as possible. In practice, however, volunteer computing is used entirely for high-throughput computing:

i.e. the goal is to maximize the rate of job completion,

not to minimize the turnaround time of a job or group of jobs.

Authors’ reply:

Thank you very much for your advice. Yes, volunteer computing is used entirely for high-throughput computing and the goal is to maximize the rate of job completion. Although BOINC currently has no direct support for directed acyclic graphs (DAGs) of dependent jobs, minimizing the makespan may be as a mechanism added to BOINC to deal with the straggler problem as you mentioned. 

Q3. By the way, some work has been done in minimizing the makespan of batches of jobs in the context of volunteer computing,

i.e. dealing with the "straggler problem".

I don't see this in the references.

Authors’ reply:

Thank you very much for your advice. Yes, we rechecked the references carefully and found a relative work about minimizing the makespan according to your suggestion. We have added the related work and references in our new version. The modified paragraphs are shown as follows: 
The 2nd paragraph in Section 2.2“In this paper, we adopt a reliability model to reduce the disturbance caused by some reasons as mentioned before. In our reliability model, we associate a trust value with each VN to predict the availability time of each VN. Usually, trust value is defined as the credibility of each VN. In the context of this paper, we define the trust value as the credibility that a VN can provide the expected availability time. Moreover, the study [33] shows that the higher trust value of VN has, the closer its expected availability time and the actual availability time are, so we associate a trust value with each VN in our reliability model to reduce the disturbance and improve the performance in task assignment. ” 
The 3th paragraph in Section 2.2“The closest work to our study is two novel task scheduling algorithms proposed by Essafi et al. [34]. These two algorithms are called the HEFT-AC (HEFT with Availability Constraint, HEFT-AC) algorithm and the HEFT-ACU algorithm (HEFT AC with unavailability, HEFT-ACU). …. However, the HEFT-AC algorithm and the HEFT-ACU algorithm are mainly about the assignment of independent tasks, which are different from our work. In our work, we mainly focus on task assignment that meets task priority requirement in VCP, and the objective is to minimize the total completion time (makespan) on the basis of improving the reliability by associating trust value of each VN.” 
The added references are shown as follows:[10]Watanabe, K.; Fukushi, M.; Horiguchi, S. Optimal spot-checking to minimize the computation time in volunteer computing. 22nd IEEE International Symposium on Parallel and Distributed Processing, IPDPS 2008, Miami, Florida USA, April 14-18, 2008. IEEE, 2008.[11]Heien,E. M.; Anderson,D. P.; Hagihara,K . Computing Low Latency Batches with Unreliable Workers in Volunteer Computing Environments. Journal of Grid Computing, 2009, 7:501-518. 

Q4. The paper's 2nd algorithm assumes that tasks are arbitrarily divisible. This is not the case for most or all existing applications of volunteer computing.

Authors’ reply:

Thank you very much for your advice. Yes, it is not the case for most or all existing applications of volunteer computing. We rechecked the references carefully and found a relative work about dividing long variable-length analyses into short BOINC workunits. On this basis, we hope our assumption may be added to BOINC as a mechanism and make BOINC a more useful resource for analyses that require a relatively fast turnaround time. For clarity, we have modified the description and added a reference. The modified paragraphs are shown as follows: 
The 2nd paragraph in Section 3.1“Although VC applications currently has no direct support for dividing long variable-sized tasks into small-sized tasks, many studies have integrated VC with other distributed computing to meet their special requirement, such as the study [35], which has discussed the strategies about how to divide long-running GARLI analyses into short BOINC workunits (a unit of work in the BOINC platform). On this basis, to extend the VC, in this paper, we assume tasks of a VC application are arbitrarily divisible. ” 

Q5. In addition, the paper has some problems in its content and presentation. It uses a number of terms without precisely defining their meaning, such as "reliability", "system reliability", "trust", "system redundancy mechanism", "reputation", and others. Similarly, it refers to "problems" without saying exactly what the problem is.

Authors’ reply:

Thank you very much for your advice. Yes, this paper has some problems in its content and presentation. We have defined the reliability and trust and deleted the description of system reliability, system redundancy mechanism and reputation according to your suggestion. The modified paragraphs are shown as follows: 
The 4th paragraph in Section 1“Usually, reliability of the distributed systems is defined as the probability that task can be executed successfully [13]. In the current context for task assignment, the definition can be narrowed down to the gap between the expected availability time and actual availability time. The gap characterizes the performance of task assignment, and the smaller the gap, the better the performance of task assignment….” 
The 2nd paragraph in Section 2.2“In this paper, we adopt a reliability model to reduce the disturbance caused by some reasons as mentioned before. In our reliability model, we associate a trust value with each VN to predict the availability time of each VN. Usually, trust value is defined as the credibility of each VN. In context of this paper, we define the trust value as the credibility that a VN can provide the expected availability time.….” 

Q6. I think I got the general idea of how the 2 algorithms work, but I was unable to follow the details, and the pseudocode is not that helpful. In cases like this, it's best to give a clear, intuitive idea of how and why the algorithm works before giving details or code.

Authors’ reply:

Thank you very much for your advice. We have reviewed our two algorithms according to your suggestions and rewritten a clear idea of how and why the algorithm works before giving details or pseudocode. The modified paragraphs are shown as follows: 
Paragraphs 1 to 3 in Section 4.1“The EFTT algorithm is a task scheduling algorithm under the constraints of available resources and prerequisite task sets. It minimizes the makespan and has two phases: a calculation phase of task execution order for computing the order of task assignment and a node selection phase for selecting the most suitable VNs according to task priority.The calculation phase of task execution order: This phase calculates task priority of each task according to the DAG. In this paper, task priority is equal to the distance between a node and the root node in the DAG. Then, this phase sorts the tasks in descending order according to their task priorities, and the sort result is task execution order R. If the tasks have the same task priority, we sort the tasks according to their order of joining (i.e. first-come-first-scheduled policy).Significantly, in this phase, the task is assigned according to task priority in task set T. When task ti is assigned, all prerequisite tasks need to be completed. Otherwise task ti cannot be assigned. And the tasks that need to be redone, they will be assigned first in the next round.” 
The 1st paragraph in Section 4.2“The basic scheduling algorithm EFTT establishes a reliability model by associating a trust value with each volunteer node to improve the performance of task assignment and minimize the makespan. However, the EFFT algorithm results in some idle computing resources and a larger number of task fragments. On the basis of the EFFT algorithm, this paper proposes an improved scheduling algorithm IEFTT. The IEFTT algorithm optimizes task scheduling in two ways: task completion time and task fragment rate. Although our second algorithm, the IEFTT algorithm shown in Algorithm 2, has a calculation phase of task execution order and a node selection phase, it uses a different method for selecting most suitable VN for each selected task. ” 

Q7. In summary: I can't recommend the paper for publication in its current state. If the presentation were improved (define terms before using them, state problem clearly, motivate algorithms better) then I wouldn't object to publication, though as I said its relevance to actual volunteer computing is dubious.

I certainly don't want to discourage the authors from doing research in volunteer computing. My longer-term recommendation is that they become involved in the BOINC community of scientists and developers, and identify research problems based on actual practice; there are many such problems.

Authors’ reply:

Thank you very much for your advice. Your constructive comments and positive support are very useful for improving the content and presentation of this paper. At the same time, your suggestions give us a direction for our future research. In the future, we will be actively involved in the BOINC community of scientists and developers, and identify research problems based on actual practice.

Thanks again for your constructive advice. I hope I can learn more knowledge from you.

Reviewer 2 Report

I found the manuscript very hard to read due to poor English. The target problem is interesting. Proposed techniques are not very sophisticated, but this is perhaps a good thing given the practical nature of the problem. The major issue is the poor presentation and formatting. The text needs to be revised by a native speaker of English. Also, the authors should cite their recent paper, which is somewhat related, and discuss the differences ("Dynamic Task Scheduling Algorithm with Deadline Constraint in Heterogeneous Volunteer Computing Platforms"). 

Further comments:`````````````````````````````````````````````````````````````````````````````````````````````````````````````````````````````````````````````````````````

- "master-slave": The use of this term is a bit problematic since it may be associated with slavery. Please have a look at other options (e.g., "master/worker").

- "reliability": This term is often associated with fault tolerance and its use is very confusing in the current problem context. Do you think "availability" would be a better term to use?

- Some notation is missing in Table 1 (e.g., "t_i.cost").

- Why is the upper bound of "Range" always equal to the expected online time? A node may be online longer than what it originally claimed, no?

- It is unclear how the lower bound of "Range" is computed. For example, in Figure 6(a), how did you come up with 2.6?

- "By analyzing historical data of multiple nodes": Which data did you analyze? How did you come up with the function given in Eq. (1)?

- The definition of task priority is very simple. It is equal to the distance between a node and the root node in the DAG. There is no need for a complicated description.

- Algorithm 1: This is completely unnecessary. You can simply say "we are sorting ...". Everybody knows what sorting is.

- "the preorder task set of the task t_i should be null": I think you mean the following: "all prerequisite tasks need to be completed"

- Algorithms are very poorly typed. Please consider using LaTeX.

- "use bubble sort algorithm": Bubble sort is one of the least efficient of sorting algorithms. Why not use quicksort?

- "Even if they ...": What do you mean here?

- Figure 8: The legend and the values on the x axis are problematic. In fact, it is better to revise all figures. They are very hard to read. Again, LaTeX is your friend.

Corrections:

- "can't" --> "cannot"

- "sudden offline": "sudden disconnection"

- "mechanism is adopted" --> "mechanisms are adopted"

- "basis of reliability model" --> basis of the reliability model"

- "Because there were ... Moreover,": This should be deleted.

- "in two aspects" --> "in two ways"

- "by volunteer node" --> "by a volunteer node"

- "will not always" --> "will not be always"

- "offline of volunteer node" --> "departure of the volunteer node"

- "without rework" --> "without being executed again"

- "it's expected" --> "its expected"

- "we introduce the trust value of volunteer node" --> "we associate a trust value with each volunteer node"

- "suddenly offline": This has no meaning.

- ". Supposing that" --> ", suppose that"

- "platforms. In" --> "platform. In"

- "number of task fragment" --> "number of task fragments"

- "will takes" --> "will take"

- "preorder task" --> "prerequisite task"

- "the task t" --> "task t" (there is no need for article when you refer to a specific item).

- "node is dynamic" --> "nodes are dynamic"

- "confident interval" --> "confidence interval"

- "online durations of node" --> "online time of node"

- "constraints is" --> "constraints are"

- "leads to the idle of some computing resources" --> "results in some idle computing resources"

- "Applications2015" --> "Applications. 2015"

Author Response

The authors have very carefully followed all constructive comments from the managing editor and two reviewers, and have tried every possible effort to clean up every issue that was raised by them. We provide below a detailed account on the changes that we have made in response to every comment that the reviewers have provided. 
The authors would like to thank the managing editor and two reviewers for their constructive comments on this manuscript and positive support. 
For all corrections, we highlight them in the new manuscript.

Round 2

Reviewer 1 Report

The revisions address many of the concerns in my first review.

Reviewer 2 Report

The authors answered all the questions I raised during the review.